# Regional Characterization Study of Fatty Acids and Tocopherol in Organic Milk as a Tool for Potential Geographical Identification

**DOI:** 10.3390/foods9121743

**Published:** 2020-11-26

**Authors:** Ill-Min Chung, Yun-Ju Kim, Hee-Sung Moon, Chang Kwon, Hee-Youn Chi, Seung-Hyun Kim

**Affiliations:** Department of Crop Science, College of Sanghuh Life Science, Konkuk University, Seoul 05029, Korea; imcim@konkuk.ac.kr (I.-M.C.); bluebeing123@konkuk.ac.kr (Y.-J.K.); hsmoon7575@konkuk.ac.kr (H.-S.M.); chang794@konkuk.ac.kr (C.K.); chi1143@konkuk.ac.kr (H.-Y.C.)

**Keywords:** organic milk, geographical identification, fatty acid, tocopherol, discriminant analysis

## Abstract

Reliable geographical identification can protect producers of excellent agroproducts, and also provide reliable purchasing information to satisfy consumers. Therefore, this study aimed to evaluate the regional and monthly variation in fatty acid (FA) and tocopherol (TOC) in organic milk (OM) and develop a geographical discriminant model of OM in Korea. In this study, OM had α-TOC and showed a regional or monthly difference of 3–5%. Moreover, C16:0, C18:1 n9 *cis* + *trans*, C18:0, and C14:0 were the predominant FAs in OM, and OM mostly had higher ∑UFA, including nutritionally desirable FAs; but lower ∑SFA among four regions or in April and August (*p* < 0.001). The model prepared using stepwise discriminant analysis showed a classification accuracy of 100% for original and cross-validated sample sets. Our results have characterized regional and monthly nutritional variations of OM, thereby potentially suggesting the applicability of a reliable Korean geographical identification labeling system using nutrient compositional analysis of OM.

## 1. Introduction

A reliable authentication method of geographical identification such as a traceability of country of origin and geographical indication (GI) of agricultural products has become critical with the increasing interest in food safety or food fraud issues [1]. According to a prior case report [2], meat products were found to have the highest frequency of food fraud, accounting for 21–22% in the Czech Republic. Moreover, beverages, in particular wine product, were also appeared with frequent food deception, because accurate identification of wine geographical origin can guarantee its quality to both consumers and producers. The European Union (EU) has recognized the potential difference in the quality of agricultural products according to the region and has introduced an integrated framework for the protection of geographical indications (PGI) and designation of origin (PDO), based on the EU regulations 2081/92 and 2082/92 of 1992 [3].

Moreover, globalization and increasing complexity of the food chains have made it more crucial to ensure the safety and traceability of the quality of agricultural products from the farm to the table. The EU defines traceability as the ability to track and investigate the production, processing, and distribution of products processed from food, feed, animals, and animal-related substances, based on the EU Food Law (CES 2001)/Codex Committee (CEC 2001). For example, since 2000 producers in the EU have had to label the origin and place of slaughter for bovines (EU Regulation 1760/2000 and 653/2014). Moreover, meat from swine, sheep, goats, and poultry must additionally include indications about the member state or the third country, in which animals are processed for each stage of meat production and distribution, as well as food business operators, after 1 April 2015 (Commission Implementing Regulation EU 1337/2013).

In Korea, to actively follow the international geographical identification protection agreement (1995 WTO Agreement on Trade-Related Aspect of Intellectual Property Rights), agricultural products have been systematically assigned certifications for quality (i.e., GI, traceability, eco-friendliness), or are managed (i.e., country of origin, LMO) under the supervision of the National Agricultural Products Quality Management Service since the late 1990s. For example, by 2018, a total of 102 agricultural products and foodstuffs had been registered as GI by 2018, since the first GI registration of green tea produced in the Boseong region in 2002. Furthermore, following the arrangement between Korea and the US (effective on 1 July 2014), and between Korea and the EU (effective on 1 February 2015), certified organic products from either country may be labeled as “organic” in both countries without additional certification process under Article 25 of the Act on Promotion of Environmentally-Friendly Agriculture and Fisheries and Management of and Support for Organic Food [4].

Various analytical approaches have been applied to achieve geographical identification based on primary/secondary metabolites, elements, stable isotope ratio (SIR), and combinations thereof using mass spectrometry, spectroscopy, separation, and others since the 1980s. In addition, chemometric tools provide sufficient discriminant powers for determining GI, traceability, and the origin of various foods (i.e., beverages, honey, wine, vegetables, cereals) based on data measured using the aforementioned analytical methods [3,5,6,7].

Moreover, interest in organic products is continually increasing because of their safety and health-related features, despite the premium cost (for organic milk (OM): 1.3 to 1.6 fold higher than conventional milk). OM was responsible for approximately 0.9% of the global milk production in 2017, which is estimated at 4.3 billion USD [8]. The OM market size in Korea is relatively small; however, it has expanded greatly (approximately 20 folds from 4.4 million USD in 2008 to 88 million USD estimated in 2018) during the past decade [9].

To date, OM authenticity has been comprehensively studied and reliably established via analysis of its isotope ratios or nutrient composition (i.e., fatty acid (FA), tocopherol, or *β*-carotenes) and content [4,10,11,12,13,14,15,16]. However, OM geographical identification has rarely been investigated so far. To our knowledge, only one prior study reported the feasibility of discriminating the regional origin of OM samples in Korea, based on SIR (i.e., C, N, O, S) combined with chemometric analysis, and reported a wrong classification probability of below 5% [17]. A few previous studies have reported the geographical origin of conventional milk from Slovenia [18,19], Malaysia [20], Australasia [21], and France [22] using SIR (i.e., C, N, O, S, or Sr), element and FA analysis, or combinations thereof.

Therefore, this study aimed to investigate FA and *α*–tocopherol (*α*–TOC) composition/content in the OM samples produced in four representative regions in Korea and develop a discriminant model to confirm the geographical identification of OM produced in Korea. Hence, this study can potentially contribute to the fair trade of OM within a country or between countries and protect the premium market recognition and the cost of OM. It can also provide sufficient purchasing information to consumers, as they have the right to know about any fraud or mislabeling.

## 2. Materials and Methods

### 2.1. Solvents and Chemicals

Ascorbic acid was obtained from Sanchun Chemical Co. (Gyeonggi-Do, Korea); sulfuric acid (H_2_SO_4_) and sodium sulfate anhydrous were purchased from Daejung Chemical and Materials Co., Ltd. (Gyeonggi-Do, Korea); potassium hydroxide (KOH) was obtained from Junsei (Tokyo, Japan). Genuine chemical standards (STDs), such as TOC homologs (*α*-, *β*-, *γ*-, *δ*-forms), 37 standard fatty acid methyl esters mixture (FAME, CRM47885), and pentadecanoic acid (C15:0, P6125) were obtained from Sigma-Aldrich, Korea. Analytical or high-performance liquid chromatography-grade solvents were used for extraction and derivatization of TOCs and FA in milk samples of interest. Methanol, ethanol, and isooctane were obtained from Fisher Scientific Korea, Ltd. (Seoul, Korea). Benzene and heptane were purchased from Junsei (Tokyo, Japan), hexane was obtained from J.T. Baker (Phillipsburg, NJ, USA), and 2,2-dimethoxypropane (DMP) and dichloromethane were procured from Sigma-Aldrich Korea (Seoul, Korea).

### 2.2. Organic Milk (OM) Sampling

OM samples of interest were monthly purchased from Boryeong, Gochang, Jeju, and Hoengseong in Korea from April 2018 to August 2018. Table 1 describes some geographical and climatic information of these four regions. Five replicates (1 L each) of homogenized/pasteurized OM samples were obtained from each region in the last week of every month (n = 5 samples/month/region). In this study, the details about the breed, age, feed regimes, and productive performance were not described for each region. However, “Holstein” is well known as the main breed at most dairy farms in Korea. Unlike the integrated milk production system, the OM dairy system in Korea typically applies more grazing pasture or fresh pasture, however it limits the use of the concentrate, total mixed ration (TMR), and some supplementation like cereals. In addition, OM farmers use conserved silage, mostly composed of rice straw organically produced during the winter season when the access to sward areas like fresh grass or clover was limited [16]. All milk samples examined in this study had already been certified and labeled as “organic” by an inspectional body, designated by the National Agricultural Products Quality Management Service, after screening for growth hormones, residual pesticides, antibiotics, and other additives. Further detailed information on OM sampling including geographical features has been previously described [16]. OM samples were frozen (–70 °C) and lyophilized (–40 °C, ~5 days). The lyophilized OM samples were promptly used for chemical analysis; then, the rest of the samples were sealed tightly and stored in a freezer (–70 °C) until further analysis.

### 2.3. Extraction and Saponification of Tocopherol (TOC) Homologs in OM

TOC homologs (*α*-, *β*-, *γ*-, *δ*-) in OM samples were extracted and saponified using a previously described method [4]. Precisely 1 g of OM powder, 0.1 g of ascorbic acid, and 20 mL of ethanol were respectively transferred to the tube and gently shaken in a water bath (80 °C, 10 min). Next, hexane and water (10 mL each) were added to the sample aliquot, which was cooled on crushed ice for 15 min. Samples were subsequently centrifuged (4 °C, ~2000× *g*, 5 min), and then the supernatant (hexane layer) was collected. Next, 10 mL hexane was added to the remaining aliquot, which was centrifuged again, and the hexane layer was collected once more. This process was repeated three times; all the hexane layers collected were washed twice with 10 mL distilled water and centrifuged (4 °C, ~2200× *g*, 10 min); then, the water layer was discarded. Lastly, the hexane layer, which was filtered through a pad of anhydrous sodium, was concentrated entirely using a vacuum rotary evaporator at 35 °C. The final residue was reconstituted with 1 mL isooctane, then transferred to an amber vial following a syringe filtering process. 

### 2.4. Instrumental Analysis of TOC Homologs in OM

TOC homologs were analyzed using an Agilent 7890B gas chromatography instrument coupled to a flame ionization detector (GC-FID) and separated via a capillary column (CP-Sil 8 CB, 50 m × 0.32 mm, 0.25 μm, Agilent Co., Ltd., Santa Clara, CA, USA). Exactly 1 μL sample aliquot was injected in the split mode (1:20). The GC-FID was set as follows: the carrier gas was N_2_, (25 mL min-1), flame gas was H_2_, (25 mL min^−1^; air, 400 mL min^−1^), and the inlet and detector temperatures were both set to 290 °C. The initial oven temperature was set to 220 °C for 2 min and programmed as follows: Increase to 290 °C (5 °C min^−1^); maintain at 290 °C (14 min); increase to 300 °C (10 °C min^−1^); finally, maintain at 300 °C (10 min). TOC STDs were prepared in isooctane at a concentration of 1000 μg mL^−1^ as a stock solution, and then appropriately diluted for the samples. In this study, α-TOC was only measured in OM; therefore, the calibration curve was prepared at a concentration range of 1−200 μg mL^−1^, showing good linearity (*r*^2^ = 0.99). *α*-TOC concentration in the samples was determined by comparing the retention time of the samples with that of the authentic STD mixtures. TOC homologs were identified by comparing the retention times of the unknown samples with those of the genuine STDs. Furthermore, genuine STDs, at proper concentrations against unknown samples, were added to the sample aliquots (sample aliquot + *α*-TOC) to further confirm the correct peak assignment for unknown OM samples [4].

### 2.5. Sample Preparation for Fatty Acid (FA) Analysis in OM

FAs in OM samples were extracted and derivatized to FAMEs for GC-FID analysis, as previously reported [4,16]. OM powder (50 mg) and an internal standard (0.2 mg, C15:0) were transferred to Teflon™-lined cap tubes, respectively. Thereafter, heptane (200 µL), along with a methylation solvent mixture (340 µL, methanol:benzene:DMP:H_2_SO_4_ = 39:20:5:2, by volume), was added for simultaneous FA extraction and FAME conversion in OM samples. The resulting mixture was gently shaken in a water bath (80 °C, 2 h). The final sample aliquot was cooled to the range of 22 °C to 25 °C and centrifuged at ~45× g for 2 min. The supernatant was used for FAME analysis.

### 2.6. Fatty Acid Methyl Esters (FAME) Analysis Using Gas Chromatography Instrument Coupled to a Flame Ionization Detector (GC-FID)

A total of 37 FAMEs in OM samples were analyzed using the same GC-FID system described above and separated via a capillary column (DB-Wax, 0.25 mm × 30 m, 0.25 μm, Agilent Co., Ltd., USA). The analytical conditions of GC-FID were set as follows: A carrier gas (helium, 35 mL min^−1^); FID flame gas (H_2_, 35 mL min^−1^; air, 300 mL min^−1^); the inlet (250 °C); FID detector (280 °C). The initial GC oven temperature was set at 50 °C for 1 min, then increased to 200 °C (25 °C min^−1^) where it was maintained isothermally for 5 min. Next, it was continually increased to 230 °C (3 °C min^−1^) and finally isothermally maintained for 20 min. The final 1-μL sample aliquot was injected into the GC-FID in the 1:20 split mode and was analyzed for 42 min per sample. FAME identification in unknown samples was achieved by comparing their retention time to that of the 37 FAME certified reference material (CRM), prepared at a 10-fold dilution in dichloromethane. Moreover, the fortified (or spiked) sample with the genuine individual FAME standard was further applied to confirm some minor or ambiguous FAMEs in OM samples. In this study, each milk FA concentration was first calculated in g kg^−1^ of dried milk powder, considering the conversion and response factor of each FA, and was finally expressed in g kg^−1^ of total FA in milk samples as follows [4]:

Fatty acid content (mg∙g^−1^, dry weight base) =
$$\frac{\;\mathrm{FA}\;i_{\;\mathrm{peak}\;\mathrm{area}\;\mathrm{in}\;\mathrm{sample}}\;\times \;\mathrm{FA}\;i_{\;\mathrm{conversion}\;\mathrm{factor}\;}\times {\;\mathrm{IS}}_{\;\mathrm{conversion}\;\mathrm{factor}}\times {\mathrm{IS}}_{\;\mathrm{amount}}}{{\mathrm{IS}}_{\;\mathrm{peak}\;\mathrm{area}\;\mathrm{in}\;\mathrm{sample}}\;\times \;\mathrm{FA}i_{\;\mathrm{response}\;\mathrm{factor}}\;\times \;\mathrm{sample}\;\mathrm{amount}}$$

FA *i*
_response factor_ =
$$\;\frac{\mathrm{FA}\;i_{\;\mathrm{peak}\;\mathrm{are}\;\mathrm{in}\;\mathrm{the}\;\mathrm{standard}\;\mathrm{mixture}}}{{\mathrm{IS}\;}_{\mathrm{peak}\;\mathrm{area}\;\mathrm{in}\;\mathrm{the}\;\mathrm{standard}\;\mathrm{mixture}}}\;\times \;\frac{{\mathrm{IS}\;}_{\mathrm{amount}\;\mathrm{in}\;\mathrm{the}\;\mathrm{FAME}\;\mathrm{STD}\;\mathrm{mixture}}}{\mathrm{FA}\;i_{\;\mathrm{amount}\;\mathrm{in}\;\mathrm{the}\;\mathrm{FAME}\;\mathrm{STD}\;\mathrm{mixture}}}$$

### 2.7. Statistical Analysis

Statistical analysis was performed with the general linear model and two-way analysis of variance (ANOVA) using SAS (version 9.3; SAS Institute Inc., Cary, NC, USA). The results of *α*-TOC and FAs measurements in OM samples were reported as mean ± standard deviation. The least significant difference test was performed at a significance level of 5%. Additionally, all *α*-TOC and FA data acquired from this study were subjected to discriminant analysis (DA) to build a predictive model for OM regional origin and production month in Korea (IBM SPSS statistics version 24, Armonk, NY, USA). In this study, for equal-sized grouping variables (four origins and five production months), all measured independents were applied to the stepwise DA (SDA) with Wilks’s Lamda method, due to the non-equivalence of covariance metrics. The predictive model developed in this study is composed of a set of discriminant functions based on linear combinations of the predictor variables showing the best discrimination between the groups. In addition, the model developed was cross-validated for unknown group membership using the leave-one-out classification, evaluated for the accuracy of group membership classification.

## 3. Results and Discussion

### 3.1. Regional and Monthly Variations of TOC and FA in OM

Table 2 summarizes the regional and monthly compositional variations of α-TOC and FA in OM samples produced from April to August 2018. *α*-TOC was found in all OM samples, and the mean α-TOC content showed a regional difference, which was 3–5% higher in Hoengseong (685.9 μg g^−1^) than in other regions (648.7 to 664.3 μg g^−1^, *p* < 0.001). Further, the mean *α*-TOC was also affected by the month (*p* < 0.01), and it showed a maximum of 4% increase in April and August compared with May to July (Table 2).

The impact of the region or month was found on the total FA content, with larger amounts of total FA in Hongseong and Jeju, and in April and August, compared to other regions and months (*p* < 0.001). Besides, Boryeong, Hongseong, and Jeju were characterized with a higher ∑UFA, especially ∑MUFA, but lower ∑SFA compared to Gochang (*p* < 0.001). We also observed a higher ∑MUFA but lower ∑SFA in April and August compared with that in May to July (Table 2).

Most FAs found in OM samples, except for C6:0, C17:1 n-7 *cis*, C22:2 n-6 *all cis*, and C23:0, were significantly different between regions, production months, and the combination of these (Table 1, *p* < 0.05). C16:0, C18:1 n-9 *cis*+*trans*, C18:0, and C14:0 were found as the predominant FAs present in OM samples, accounting for approximately 80% of the total FA amount, which showed a regional difference but no trend (Table 2, Figure 1). In particular, C16:0 in Hoengseong and C18:1 n-9 *ci s*+ *trans* in Jeju were observed as outliers and extremes due to the large monthly variations (Figure 1).

C18:2 n-6 *all cis*, C12:0, and C10:0 were the next most abundant FAs in OM samples, and their concentration also differed according to the region and the production month (Table 2, *p* < 0.001). In particular, compared to those from other regions, OM samples from Boryeong contained higher concentrations of essential FAs, like C18:2 n-6 *all cis* and C18:3 n-3 *all cis* (Table 2, Figure 1, *p* < 0.001). Besides, C18:2 n-6 *all cis* and C18:3 n-3 *all cis* concentrations were also higher in July and August than in May and June (Table 2). Interestingly, despite lower total FA content in Boryeong, ∑n-3, ∑n-6, and the ratio of n-3/n-6, known as nutritionally desirable FA factors, were higher in OM samples from Boryeong, than in those from Hoengseong and Jeju with a higher total FA content (Table 2, Figure 1). Notably, most FA contents significantly differed across different months (*p* < 0.001), being higher in July and August than in May and June (Table 2). Finally, long-chain FAs, notably C13 to C21, accounted for ~88% of the total FA content in OM samples of interest (Table 2), similar to our previous findings [4,16].

Furthermore, hierarchical cluster correlation analysis was performed to investigate significant relationships among 34 variables. Among four clusters, the first two clusters (blue and green) included mostly short/medium-chain saturated FAs, and the third cluster (orange) contained a mix of long-chain (un) saturated FAs and α-TOC. The last cluster (red) contained the rest of the FAs, including long-chain/polyunsaturated FAs (Figure 2). Strong positive correlations (*r* ≥ 0.93) were revealed among C6:0, C8:0, C10:0, or C14:0, while negative correlations (*r* = –0.95) were observed between C14:1 n-5 *cis* and C18:0 (Figure 2). 

Despite debates on the nutritional quality associated with health benefits between organic and conventional products, so far, nutritional parameters such as FA, TOC, and mineral content in OM have been investigated comprehensively and are well documented [12,23,24]. In particular, different feeding regimes between organic and integrated farm systems are widely known as main contributors to the production of milk with high nutritional value [11,13,25].

According to systematic reviews [12,23,24], the feeding regime composed of more grazing, roughage, or conserved forage, which are typically applied in OM production, is the reason why OM is more nutritionally valuable than conventional milk. For example, OM contains more nutritionally favorable FAs, as well as a higher *α*-TOC content, compared to conventional milk produced in an integrated farm system. In particular, high PUFA and n-3 PUFA (especially, *α*–linolenic acid), but low ∑n-6:∑n-3 ratio, is reported in OM. A recent comprehensive review also reported the practical/external influences such as feeding regimes were more critical factors on FA profile and content in milk, and its qualified estimate was ~55%; while, the effect of cow breed appeared as a minor factor on milk FAs profile, which was responsible for ~20% [26]. Furthermore, fresh forage increases 3R *α*-TOC isomers in milk, while increased intake of concentrate, corn silage, other silage, hay, and straw decreases *α*-TOC content in milk [11,12]. These are also similar to our current findings on the nutritional features of OM in Korea.

Moreover, because of increased possibility of outdoor feeding with fresh grass or forage during the summer, *α*-TOC and FA content in milk are usually higher in the summer than in winter [11,27]. The current study also showed a monthly variation in α-TOC and FA content in OM, which is consistent with previously reported findings in Korea [4]. However, the seasonal effect on *α*-TOC and FA content in OM was not apparent in the current study, indicating these contents were higher in August (summer) than in May to July, but similar to that in April (spring). Different feeding regimes (i.e., forage type and intensity, forage conservation method, concentration, TMR), considering geographical and climatic features of four OM producing regions, may partly describe the weak/unclear seasonal variation of *α*-TOC and FA content in OM in this study. Thus, using the well-defined OM nutritional data from four regions reported in this study, feasibility of the OM geographical identification discriminant model in Korea could be examined using SDA.

### 3.2. Stepwise Discriminant Analysis of OM Regional Origin and Production Month

SDA was used to predict OM classification to establish OM geographical identification in Korea. SDA was conducted using the dataset comprising all OM samples (from four regions), and only 16 variables among a total of 34 variables (FAs, *α*-TOC) were used to develop the model using the SDA method in this study.

The OM samples were distinguished among four regions in Korea using the SDA approach (Figure 3a). The first function alone was explained by 73.5% of the total variance, and the cutting score value was −6.25, which was clustered between Jeju and Hoengseong (Figure 4a). Furthermore, the first function showed high eigenvalue (99.215) and canonical correlation (*r* = 0.995), which suggested that the model explained 99.0% of the variation in the grouping of the OM production region. The second function was described by 16.5% of the total variance, with an eigenvalue of 22.324, and canonical correlation (*r* = 0.978) in this model. The cutting score values were −3.30 for Hoeongseong and 3.22 for Boryeong from the cluster of Gochang and Jeju (Figure 3a and Figure 4b).

The Wilks’ lambda value (0.0000) indicates that the three discriminant functions were highly significant (*p* = 0.0000), meaning there was 0% unexplained (see Appendix A). The major contributors for discriminating OM from four different regions in Korea were C10:0 (standardized coefficient, 7.187), C8:0 (−4.243), C12:0 (−4.121), and C6:0 (3.093) in the first function, and C8:0 (10.852), C10:0 (−9.421), C14:0 (4.469), and C12:0 (−4.206) in the second function (see Appendix A). Hence, this model showed a classification accuracy of 100% to the original sample set; furthermore, the cross-validation using the leave-one-out method for unknown samples also showed 100% classification accuracy (Table 3).

Moreover, we developed a discriminant model for OM production months, and the first two functions were described as 75.4% of the total variance (Figure 3b). The first two cutting score values did not discriminate OM production months for June vs. August (Appendix A) and April vs. July (Appendix A). However, the next two functions (3rd, 4th) described clear discriminations of the above unclear clustering of production months (Appendix A). The first and second functions for grouping OM production months had eigenvalues of 12.228 and 5.472, respectively, suggesting that this discriminant model explained 92.4% and 84.6% of the variations in the grouping, respectively (see Appendix A). Further, the first function, along with C10:0 (−7.179) and C14:0 (3.691), and the second function with C8:0 (10.852) and C14:0 (4.469) were described as the most important parameters of the model for grouping variables (see Appendix A). Thus, OM production months, using SDA, had classification accuracies of 98.0% and 95.0% in the original group and the cross-validated group, respectively (Appendix A).

In general, DA is similar to multiple linear regressions, based on the prediction of an outcome, and it minimizes the possibility of misclassifying cases into respective groups or categories. A linear discriminant analysis (LDA), in particular, is a common supervised pattern recognition method to maximize the ratio of inter-class variance and also minimize that of intra-class variance [28]. Prior studies have reported the suitability of this chemometric approach to determine the authenticity, including the geographical origin and organic status of various agriculture products and foodstuffs [28,29,30].

However, the LDA is typically hard to apply when the covariance matrix is a singular or abnormal distribution of data in groups. Hence, if the covariance matrix differs in at least two groups based on the evaluation by the Box’s M test, quadratic (QDA), canonical (CDA), or stepwise DA (SDA) has to be used to develop discriminant models [31,32]. In this study, the log determinants appear similar, while the Box’s M was 2390.388 with an F of 4.064, which was significant at *p* < 0.000 (data not shown), so a null hypothesis of equality of covariance matrices was rejected. However, given the large sample, it is not usually considered serious.

According to previous studies, the geographical identification of conventional milk produced in Southern Italy, France, and Slovenia was determined by analyzing SIR, FA, elements, or organic constituents, combined with multivariate analyses such as principal component analysis (PCA), LDA, or SDA. For example, the PCA model obtained using SIR, elemental, and nuclear magnetic resonance (NMR) data enables the differentiation of milk samples in Southern Italy (Apulian) from foreign (Central Europe) samples, where the first principle component is highly critical. Furthermore, its classification and prediction abilities by DA are 95% and 90% using samples of known geographical origin [33]. The δ^18^O is described as a critical parameter to differentiate the milk production sites in France (mountain vs. plain) based on DA [22]. Moreover, the function 1 (Cl, Zn, P, Ca) and 2 (K, Cl, P, Zn) of the discriminant model by DA explain 91.2% of the total variances and enable the differentiation of milk in Pannonian and Mediterranean regions from that in other regions, like Alpine and Dinaric in Slovenia [18]. Recently, the overall temporal predication variability using SIRs and element data combined with DA method showed 84.6% and 56.4% for regional differences (i.e., Mediterranean, Pannonian, Alpine, Dinaric) of Slovenian cow milk samples collected for 2012 to 2014. In particular, the prediction ability appeared the highest (82.1%) for the Pannonian and lowest (26.9%) for Alpine region [19].

Meanwhile, a recent study [17] has reported the feasibility of GI of the OM samples in Korea using SIR analysis combined with orthogonal partial least squares (OPLS)-DA. The model by OPLS-DA shows that the geographical origin of OM samples in Boryeong, Gochang, and Jeju can accurately be predicted with a probability of wrong classification below 5%, based on the external validation; however, OM samples in Hoengseong show a high probability (7.8–23.8%) of wrong classification compared to the other three regions, typically indicated as unacceptable in food analysis. Therefore, the present study was conducted to develop and compare another discriminant model of nutrient analysis combined with SDA, applied to the same OM samples as the previously described study. The current SDA model of nutrient data showed a classification accuracy of 100% for both original samples and the cross-validation set, and it showed a better discriminant and predicting power for all four regions (Boryeong, Gochang, Jeju, Hoengseong) examined in this study, compared to the previously reported OPLS-DA model based on SIR analysis [17].

## 4. Conclusions

This study shows a milk nutrient analysis-based SDA approach (similar to *α*-TOC and FA) to develop OM geographical identification in Korea. *α*-TOC and FA contents in OM samples vary significantly according to the region, production month, and interactions thereof.

The SDA models showed a classification accuracy of 100% for OM geographical identification for both original and cross-validation sample sets, with C10:0, C8:0, and C14:0 appearing as major contributors. Hence, the OM nutrient-based SDA model showed an excellent discriminative and predicting power for OM geographical identification in Korea, which is comparable to other chemometric models based on SIR, metabolites, or elements.

In conclusion, this study will enhance the premium market recognition and cost of genuine OM products, and enable customers to avoid fraudulent or mislabeled OM samples. However, further studies are needed to validate monthly/seasonal variation of *α*-TOC and FA contents in OM produced in four regions in Korea, to delineate regional differences or traceability from the farm to the consumer’s table.

## Figures and Tables

**Figure 1 foods-09-01743-f001:**
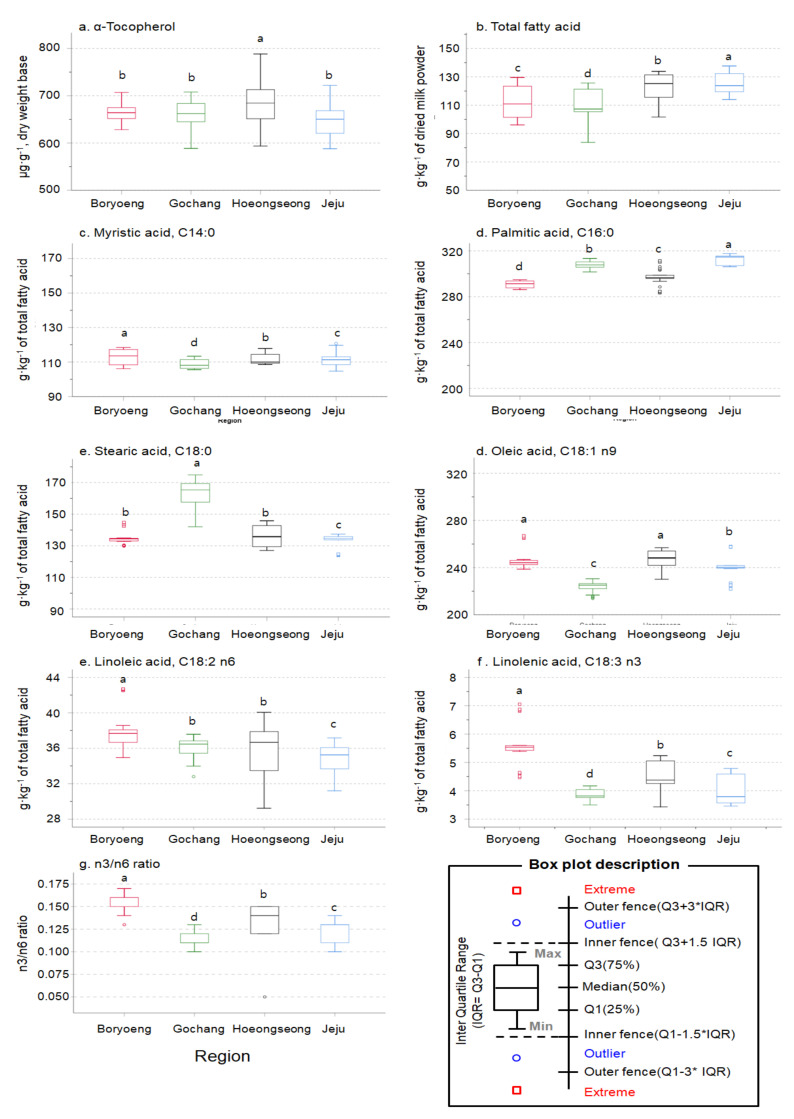
Regional differences of α-tocopherol (α-TOC) and the selected representative fatty acid (FAs) contents in organic milk (OM, n = 25 per each region, *p* < 0.05).e. Linoleic acid, C18:2 n6 f. Linolenic acid, C18:3 n3 g. n3/n6 ratio.

**Figure 2 foods-09-01743-f002:**
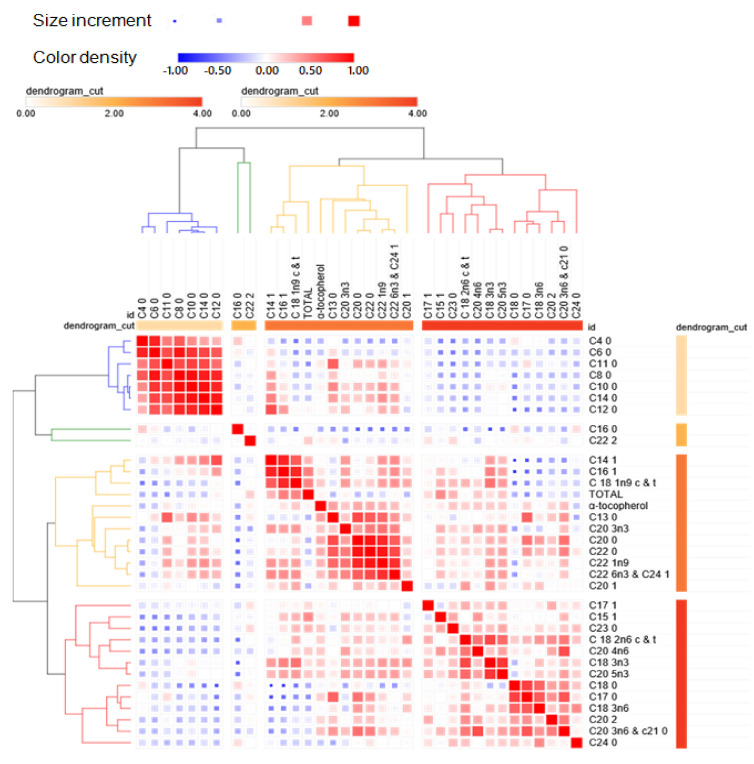
Hierarchical cluster correlation matrix of *α*-tocopherol (*α*-TOC) and fatty acids (FAs) in organic milk (OM). Each square indicates the Pearson’s correlation coefficient of a pair of compounds, and the correlation coefficient values are represented by the intensity of blue or red color combined with the square size, as indicated on the color and size scale.

**Figure 3 foods-09-01743-f003:**
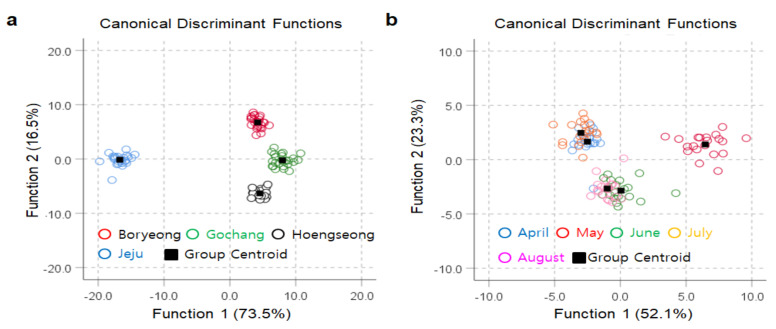
Stepwise discriminant analysis (SDA) for the regional authenticity (**a**) and organic milk (OM) production month (**b**) using α-tocopherol (α-TOC) and fatty acids (FAs) measured in OM.

**Figure 4 foods-09-01743-f004:**
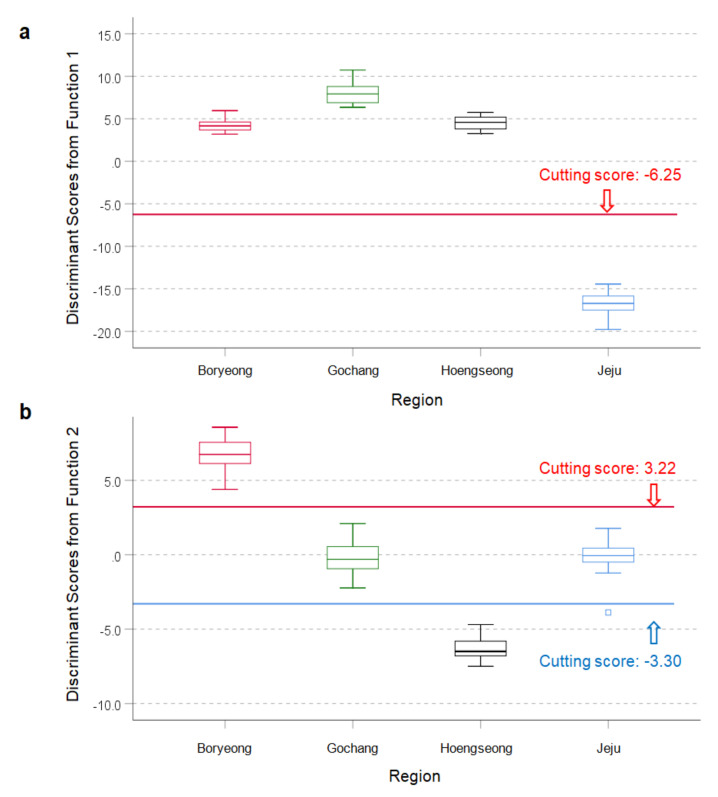
Box-and-whisker plots for discriminant scores from the function 1 (**a**) and the function 2 (**b**) of the discriminative model (see Figure 3a) developed for organic milk (OM) regional authenticity (see Figure 1. about the description for box-and –whisker plot).

**Table 1 foods-09-01743-t001:** Brief descriptions of geographical location, features, and climatic factors of four representative organic milk (OM) production regions in Korea.

Region	Location	Geographical Features	Distance from the Nearest Coast		April	May	June	July	August
Boryeong	36° 51′N, 126° 53′E, ~50m a.s.l.	open ocean, near coast area	~4.3 km	Mean temperature, °C	12.1	17.2	21.4	26.7	27.6
				Total precipitation, mm	128.1	104.5	71	262.7	239.6
Gochang	35° 45′N, 126° 45′E, ~20 m a.s.l.	open ocean, near coast area	~1.5 km	Mean temperature, °C	13	17.6	21.7	26.6	27.3
				Total precipitation, mm	102	64.2	168.4	183.7	269.6
Jeju	33° 35′N, 126° 33′E, ~350 m a.s.l.	island, near coast, mountain area	~10 km	Mean temperature, °C	15.9	19.1	22.3	27.3	28.8
				Total precipitation, mm	112.5	98.8	211.1	48.7	376.5
Hoengseong	37° 46′N, 128° 09′E,~ 320 m a.s.l.	inland, mountain area	~81 km	Mean temperature, °C	12	17.1	22.6	26.6	27.2
				Total precipitation, mm	143.8	268.6	101.4	204.5	300.6

**Table 2 foods-09-01743-t002:** Means of regional and monthly difference of α-tocopherol (α-TOC, µg·g^−1^, dry weight base) and fatty acid content (FA, g·kg^−1^ of total fatty acids) in organic milk (OR).

	Region (R), *n* = 25 Per Each Region					Milk Production Month (M), *n* = 20 Per Each Month						*p*-Value		
												Main Factor		Interaction
	Boryeong	Gochang	Hoengseong	Jeju	LSD_0.05_	APR	MAY	JUN	JUL	AUG	LSD_0.05_	R	M	R*M
α-tocopherol, µg·g^−1^, dry weight base	664.3	660.3	685.9	648.7	16.0	681.5	651.2	665.4	654.1	671.9	17.9	***	**	**
C4:0	21.9	23.0	21.9	22.5	0.47	21.4	25.2	20.9	22.7	21.5	0.52	***	***	***
C6:0	17.8	17.5	17.7	17.8	0.26	17.4	20.1	16.9	17.6	16.4	0.30	ns	***	***
C8:0	11.2	10.5	11.1	11.0	0.15	11.2	12.3	10.6	10.7	10.1	0.17	***	***	***
C10:0	25.9	24.0	26.0	24.5	0.33	26.1	27.9	24.7	24.1	22.8	0.37	***	***	***
C11:0	0.55	0.53	0.57	0.46	0.03	0.52	0.72	0.51	0.46	0.43	0.04	***	***	***
C12:0	31.9	29.0	32.5	33.0	0.24	32.8	34.6	31.5	30.1	29.0	0.27	***	***	***
C13:0	0.95	0.98	1.04	0.78	0.01	0.98	1.00	0.94	0.88	0.90	0.01	***	***	***
C14:0	112.7	108.9	111.9	111.3	0.53	112.6	116.7	111.8	108.1	106.7	0.59	***	***	***
C14:1 n-5 *cis*	9.68	6.73	8.94	9.67	0.08	9.17	9.10	8.91	8.12	8.48	0.09	***	***	***
C15:1 n-5 *cis*	0.94	1.00	1.23	0.99	0.18	1.43	0.42	1.31	0.93	1.10	0.20	**	***	***
C16:0	290.9	307.7	296.6	312.2	0.68	299.8	304.6	303.2	301.8	299.8	0.77	***	***	***
C16:1 n-7 *cis*	15.1	12.4	15.1	14.9	0.12	14.9	14.0	14.4	13.8	14.8	0.14	***	***	***
C17:0	10.6	11.8	10.4	8.8	0.04	10.3	10.2	10.5	10.5	10.5	0.05	***	***	***
C17:1 n-7 *cis*	2.62	2.49	2.43	2.54	0.26	2.68	2.29	2.38	2.58	2.68	0.29	ns	*	**
C18:0	135.2	161.8	135.8	133.2	0.57	135.2	136.2	142.1	147.9	146.1	0.64	***	***	***
C18:1 n-9 *cis*+*trans*	247.7	223.7	247.3	241.4	1.40	242.9	229.6	240.8	238.5	248.5	1.57	***	***	***
C18:2 n-6 *all cis*	38.3	36.1	36.0	34.9	0.35	36.4	33.8	35.4	38.8	37.0	0.39	***	***	***
C18:3 n-6 *all cis*	1.27	1.29	1.15	1.14	0.03	1.22	1.18	1.22	1.23	1.23	0.03	***	**	***
C18:3 n-3 *all cis*	5.61	3.88	4.55	4.03	0.07	4.81	4.04	4.28	4.72	4.74	0.08	***	***	***
C20:0	2.54	2.34	2.39	1.93	0.02	2.40	2.24	2.47	2.19	2.21	0.02	***	***	***
C20:1 n-9 *cis*	4.01	3.39	3.53	3.04	0.13	3.21	3.58	3.43	3.62	3.64	0.15	***	***	***
C20:2 n-6 *all cis*	0.57	0.63	0.60	0.53	0.04	0.60	0.49	0.59	0.67	0.56	0.05	***	***	***
C20:3 n-6 *all cis*	1.83	1.89	1.84	1.65	0.03	1.83	1.67	1.86	1.92	1.73	0.03	***	***	***
C20:4 n-6 *all cis*	2.09	2.09	2.15	2.08	0.05	2.15	1.91	2.14	2.26	2.04	0.05	*	***	***
C20:3 n-3 *all cis*	0.27	0.18	0.27	0.23	0.03	0.24	0.21	0.29	0.23	0.21	0.04	***	***	*
C20:5 n-3 *all cis*	0.55	0.42	0.47	0.42	0.03	0.52	0.40	0.46	0.48	0.46	0.03	***	***	***
C22:0	1.39	1.22	1.27	1.04	0.02	1.32	1.17	1.38	1.14	1.15	0.02	***	***	***
C22:1 n-9 *cis*	1.60	0.43	1.08	0.20	0.06	1.15	0.78	1.35	0.29	0.55	0.07	***	***	***
C22:2 n-6 *all cis*	1.40	1.67	1.50	1.44	0.35	2.03	1.41	0.84	1.19	2.03	0.39	ns	***	ns
C23:0	0.75	0.72	0.71	0.72	0.06	0.80	0.56	0.80	0.66	0.82	0.07	ns	***	**
C24:0	1.63	1.47	0.33	1.35	0.08	1.29	0.94	1.37	1.21	1.17	0.09	***	***	***
C22:6 n3 *all cis*	0.38	0.22	0.33	0.23	0.04	0.34	0.26	0.36	0.23	0.26	0.04	***	***	***
Total (g·kg^−1^ of dried milk powder) ^†^	112.1	109.6	122.2	125.0	2.27	129.2	104.8	112.3	115.3	124.7	2.53	***	***	***
Calculated value, g·kg^−1^ of total fatty acids														
∑ SFA	666.1	701.5	670.3	680.6	1.76	674.1	694.6	679.7	680.1	669.7	1.97	***	***	***
∑ UFA	333.9	298.5	328.5	319.4	1.75	325.6	305.2	320.0	319.6	330.0	1.96	***	***	***
∑ MUFA	281.7	250.2	279.6	272.8	1.53	275.4	259.7	272.5	267.9	279.7	1.71	***	***	***
∑ PUFA	52.2	48.3	48.9	46.6	0.50	50.2	45.4	47.5	51.7	50.3	0.56	***	***	***
PUFA/MUFA	0.19	0.19	0.18	0.17	0.003	0.18	0.18	0.17	0.19	0.18	0.003	***	***	***
∑ n-6 PUFA	43.4	41.3	41.2	39.8	0.41	41.6	38.6	40.7	44.2	42.0	0.46	***	***	***
∑ n-3 PUFA	6.80	4.71	5.52	4.91	0.20	5.91	4.77	5.40	5.66	5.68	0.22	***	***	***
n3/n6	0.16	0.11	0.13	0.12	0.01	0.14	0.12	0.13	0.13	0.14	0.01	***	***	***
∑ Short/Medium (4–12)	109.4	104.5	109.8	109.3	1.32	109.4	120.9	105.1	105.6	100.3	1.47	***	***	***
∑ Long (13–21)	883.4	889.8	883.7	885.7	1.36	883.3	873.7	888.5	889.3	893.4	1.52	***	***	***
∑ Very long (≥ 22)	7.15	5.73	5.23	4.98	0.38	6.94	5.13	6.09	4.73	5.99	0.43	***	***	***

^†^Total fatty acid indicates the sum of each fatty acid measured in this study, and is only expressed as g·kg^−1^ of dried milk powder. Saturated fatty acid (SFA), Unsaturated fatty acids (UFA), Monounsaturated fatty acid (MUFA), Polyunsaturated fatty acid (PUFA), omega-6 fatty acid (n-6), omega-3 fatty acid (n-3), Short and medium chain fatty acids (Short/Medium, including 4 to 12 carbon fatty acids), Long chain fatty acids (Long, including 13 to 21 carbon fatty acids), Very long-chain fatty acids (Very long, including more than 22 carbon fatty acids), ns: non-significant, *: *p* < 0.05, **: *p* < 0.01, ***: *p* < 0.001.

**Table 3 foods-09-01743-t003:** Classification and cross-validated results ^a,b^ of organic milk (OM) geographical identification in Korea by stepwise discriminant analysis (SDA).

			Predicted Group Membership				Total
			Boryeong	Gochang	Hoengseong	Jeju	
Original	Count	Boryeong	25	0	0	0	25
		Gochang	0	25	0	0	25
		Hoengseong	0	0	25	0	25
		Jeju	0	0	0	25	25
	%	Boryeong	100	0	0	0	100
		Gochang	0	100	0	0	100
		Hoengseong	0	0	100	0	100
		Jeju	0	0	0	100	100
Cross-Validated ^c^	Count	Boryeong	25	0	0	0	25
		Gochang	0	25	0	0	25
		Hoengseong	0	0	25	0	25
		Jeju	0	0	0	25	25
	%	Boryeong	100	0	0	0	100
		Gochang	0	100	0	0	100
		Hoengseong	0	0	100	0	100
		Jeju	0	0	0	100	100

^a^ 100.0% of original grouped cases correctly classified. ^b^ 100.0% of cross-validated grouped cases correctly classified. ^c^ Cross validation is undertaken only for those cases in the analysis. In cross validation, each case is classified by the functions derived from all cases other than that case.

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
