# Peer review of "Regional Characterization Study of Fatty Acids and Tocopherol in Organic Milk as a Tool for Potential Geographical Identification"

_foods, 2020, doi:10.3390/foods9121743_

Round 1

Reviewer 1 Report

The fatty acids and tocopherol in organic Milk were analyzed in samples collected from different regions of South Korea.  The analytical methods are valid and the results are meaningful to consumers.  However, I am not sure what are the differences between the 4 regions geographically and feed regimes although the locations were described.  What are the reasons (hypothesis) for the samples collected from the regions monthly? The information of the cows need to be reported in details such as breed, age, productive performance and management.  The potential impacts need to be discussed if the breed and age are different. Are there any CLA and trans FA in the OM ? The method for calculation of the FA (g.kg-1) need to be reported. Is there any potential impact of C15:0 on the results?  Is the C4:0 and C6:0 content actuate using the trans-methylation methods?  

Reviewer 2 Report

Comments and recommendation

29-30: I can´t see the connection between this information (the occurrence of bovine spongiform encephalopathy (often called mad-cow disease), foot and mouth disease, and chicken influenza) and GI; … please remove it …

I think more important when regarding the topic of manuscript is an authenticity of food. I recommend to authors to add some information about food fraud -  e.g. Koubová et al., 2018, Molecules; …

69-71: Please consider the comprehensive review on feed factors  - e.g. Hanuš et al., 2018, Molecules; …

to state abbreviation correctly: for example l. 31 – EU, l. 70 – FAs, tocopherol; 

to explain abbreviation also in captions of tables and figures

190-191: … in April and August compared with May to July (Table 1, p < 0.01) – this information (significance level) is not in the Table 1, there is only the effect of month; similarly: 211-213: … C18:2n-6 and C18:3n-3 concentrations were also higher in July and August than in May and June 212 (Table 1, p < 0.0001); 216-217 …

Therefore, I suggest indicate the significance between the groups (using by different superscripts) in Table 1

Table 1. Summary of regional and monthly difference … it would be better the word means

193-194, 209, 280 (for example) and Table 1: In research publications, it is not usual to express the significance as p < 0.0001 (it is usually used p < 0.001 (***) for the highest significance level)

to unify P (Table 1, Figure 1) or p (text)

197, 199, 201 (for example) and Table 1: please, specify isomers of monounsaturated FAs, especially C18:1

Figure 1: please, unify the y-axis (min – max values), at least for FA with high content (C14:0, C16:0, C 18:0, and C18:1 n-9) and for FA with low content (linoleic and alpha-linolenic acids) 

259-260: I cannot agree with information, that FAs C6 to C12 are medium-chain and C12 to C16 belong to long-chain FAs …

In Supplementary SDA Tables, there are many commas instead dots in the values

In some cases, the number of decimal places is too high (standardized coefficient, eigenvalues etc.)

In Conclusions, it is not usual to repeat the results, to use significant levels or to mention citation

355-361 I recommend deleting it, 362-368 I recommend reducing it

Round 2

Reviewer 1 Report

The replies satisfied with this reviewer.

Author Response

Comments and Suggestions for Authors: The replies satisfied with this reviewer.

Author response: We sincerely appreciate your kind and fast review as well as insightful comments, which are significantly improved to the scientific quality of this manuscript. Please kindly consider our response.